# The Metabolic Profiles of Metabolically Healthy Obese and Metabolically Unhealthy Obese South African Adults over 10 Years

**DOI:** 10.3390/ijerph19095061

**Published:** 2022-04-21

**Authors:** Herculina Salome Kruger, Zelda De Lange-Loots, Iolanthé Marike Kruger, Marlien Pieters

**Affiliations:** 1Centre of Excellence for Nutrition, North-West University, Potchefstroom 2520, South Africa; zelda.delange@nwu.ac.za (Z.D.L.-L.); marlien.pieters@nwu.ac.za (M.P.); 2Medical Research Council Unit for Hypertension and Cardiovascular Disease, North-West University, Potchefstroom 2520, South Africa; 3African Unit for Transdisciplinary Health Research, North-West University, Potchefstroom 2520, South Africa; lanthe.kruger@nwu.ac.za

**Keywords:** obesity, metabolic syndrome, cardiometabolic risk, Africa

## Abstract

Obesity is associated with an increased cardiometabolic risk, but some individuals maintain metabolically healthy obesity (MHO). The aims were to follow a cohort of black South African adults over a period of 10 years to determine the proportion of the group that maintained MHO over 10 years, and to compare the metabolic profiles of the metabolically healthy and metabolically unhealthy groups after the follow-up period. The participants were South African men (*n* = 275) and women (*n* = 642) from the North West province. The prevalence of obesity and the metabolic syndrome increased significantly. About half of the metabolically healthy obese (MHO) adults maintained MHO over 10 years, while 46% of the women and 43% of men became metabolically unhealthy overweight/obese (MUO) at the end of the study. The metabolic profiles of these MHO adults were similar to those of the metabolically healthy normal weight (MHNW) group in terms of most metabolic syndrome criteria, but they were more insulin resistant; their CRP, fibrinogen, and PAI-1act were higher and HDL-cholesterol was lower than the MHNW group. Although the metabolic profiles of the MUO group were less favourable than those of their counterparts, MHO is a transient state and is associated with increased cardiometabolic risk.

## 1. Introduction

The prevalence of adult obesity in South Africa is higher than in most low- and middle-income countries, with 39.2% of women and 10% of men being obese [1,2]. During the 1980s, Walker et al. referred to obesity among women living in African countries as “healthy” obesity [3]. Several recent studies confirmed that some overweight and obese individuals are not at increased cardiometabolic risk and could be described as metabolically healthy obese (MHO). In contrast, some individuals within the normal body mass index (BMI) range (18.5 to 25 kg/m^2^) have abnormal metabolic profiles [4,5,6]. Furthermore, the BMI cut-off point to differentiate between individuals with excessive versus normal body fat differs across ethnicities, so BMI may underestimate excess adiposity in some populations [7,8,9]. Despite these limitations, BMI is still the most widely used method to define overweight and obesity in epidemiological studies and is regarded as a good index of cardiometabolic risk [10]. Diet and other lifestyle exposures during early childhood may influence later cardiometabolic risk [11]. Some individuals appear to have an increased predisposition to metabolic syndrome (MetS) at a BMI below the overweight cut-off point and may therefore have a metabolically unhealthy normal weight, whereas other individuals maintain cardiometabolic health, even when overweight [4,5,7].

Scientists do not agree on an internationally accepted definition for MHO and the prevalence of MHO varies in different settings [4,5,12]. In most studies MHO is defined as the absence of the MetS in overweight and/or obese adults [4,13], although a stricter definition has been proposed as those with a BMI > 30 kg/m^2^ and having none of the MetS criteria [14,15]. The MetS criteria include three or more of the following: high blood pressure, fasting plasma glucose, triglycerides, waist circumference and low HDL-cholesterol [16]. This definition of the MetS has been criticised, as the presence of MetS does not indicate the severity of cardiovascular disease (CVD) risk [17]. However, a meta-analysis showed that individuals diagnosed with MetS are at increased risk of CVD [18].

Visceral adiposity and intrahepatic fat, rather than general obesity, are associated with insulin resistance [19]. Fasting insulin, markers of low-grade inflammation and haemostatic variables are associated with abnormal metabolic profiles, but are not included as MetS parameters, because these variables are not measured in primary care settings [17]. However, these variables are important predictors of increased risk of CVD and a more detailed definition of MHO, including markers of insulin resistance and inflammation was recently proposed [14]. A recent study among older adults revealed that those with the MetS had higher levels of oxidative stress markers, which may contribute to increased inflammation and development of CVD [20]. To our knowledge, no data regarding the metabolic profiles of MHO African populations are available. Therefore, the purpose of this study was (1) to follow a cohort of Black South African adults over a period of 10 years to determine what proportion of the group maintained the MHO state over 10 years, and (2) to compare the metabolic profiles of the MHO and metabolically unhealthy overweight/obese (MUO) groups at the end of the follow-up period. These results contribute knowledge that may assist an international working group to establish an internationally accepted definition for MHO, which may need to include more CVD risk markers than MetS alone.

## 2. Materials and Methods

### 2.1. Design, Setting and Participants

The cohort represents the South African leg of the international Prospective Urban and Rural Epidemiological (PURE) study [21]. The participants were recruited from urban and rural sites in the North West Province, South Africa. Fieldworkers visited 6000 households and compiled a list of 4000 adults older than 30 years with no acute illness. During August to November 2005, the recruited participants were transported to the university study centre where they gave informed consent for the measurements performed after overnight fasting.

From the initial list, 2010 participants were available for testing at baseline, but complete data were collected from 1195 women and 742 men. Due to loss to follow-up, 642 women and 275 men were followed up to 2015. The total number lost to follow-up (1020), represented 33% deaths, 26% relocations, 28% refusals for further participation, and 13% lost without contact information. Complete data at both baseline and end were available for 917 participants of this cohort.

### 2.2. Measurements

Sociodemographic information, anthropometric variables, blood pressure, fasting glucose, glycated haemoglobin (HbA1c), insulin, C-reactive protein (CRP), lipids, fibrinogen and plasminogen activator inhibitor-1 activity (PAI-1_act_) were measured. Trained fieldworkers used a structured sociodemographic and lifestyle questionnaire to interview the participants. Responses included age, educational status, living area, smoking, alcohol consumption, chronic diseases, including HIV, and chronic medication used.

Anthropometrists used standardised procedures and calibrated instruments to measure height, weight and waist circumference [22]. Body weight in lightweight clothing was measured to the nearest 0.1 kg on a calibrated digital scale and height was recorded to the nearest 0.1 cm using a stadiometer (Seca, Hamburg, Germany). BMI was calculated as weight (kg) divided by height (m) squared. Waist circumference was measured to the nearest 0.1 cm using a flexible steel tape (Lufkin, Cooper Tools, Apex, NC, USA) at the horizontal level between the lower rib and iliac crest. Blood pressure measurements were performed in duplicate (5 min apart) on the right upper-arm after a 10 min rest period, while the participants were seated with the right arm supported at heart level. Systolic (SBP) and diastolic blood pressure (DBP) were measured with an OMRON device (Omron Healthcare, Kyoto, Japan), using appropriately sized cuffs for obese participants.

Eight-hour fasting blood samples were collected from the antebrachial vein, using a sterile infusion set and syringes. After centrifugation, serum and plasma samples were stored in aliquots at −80 °C until analysed. Sodium fluoride glucose was measured using an enzymatic reference method with hexokinase (Vitros DT 6011 Chemistry Analyser, Ortho-Clinical Diagnostics, Rochester, NY, USA) in 2005 and Cobas Integra 400 Roche^®^ Clinical System (Roche Diagnostics, Indianapolis, IN, USA) in 2015. Total serum cholesterol, triglycerides (TG) and high-density lipoprotein (HDL)-cholesterol were measured by means of an enzymatic colorimetric method by Konelab20iTM auto-analyser (Thermo Fisher Scientific Oy, Vantaa, Finland) in 2005 and a Cobas Integra 400 Roche^®^ Clinical System (Roche Diagnostics, Indianapolis, IN, USA) in 2015. High-sensitivity CRP was measured by means of a particle enhanced turbidimetric assay. We measured HbA1c in whole blood (EDTA) samples with ion-exchange high-performance liquid chromatography using the D-10 Haemoglobin testing system from Bio-Rad (Bio-Rad Laboratories Ltd., Hercules, CA, USA). Insulin was measured by immunoassay (Insulin Elecsys, Roche Diagnostics, Indianapolis, IN, USA) and homeostatic model assessment of insulin resistance (HOMA-IR) was calculated according to the formula: fasting insulin (µU/L) × fasting glucose (mmol/L)/22.5 [23]. Fibrinogen was measured using a modified Clauss method (Multifibrin U-test, BCS analyser, Dade Behring, Deerfield, IL, USA) for the 2005 samples and an ACL-200 (Automated Coagulation Laboratory analyser, Instrumentation Laboratories, Milan, Italy) for the 2015 samples. Plasminogen activator inhibitor-1 activity (PAI-1_act_) was measured with an indirect enzymatic assay (Spectrolyze PAI-1, Trinity Biotech, Bray, Ireland). In some instances, the sample analyses were performed with different apparatus at the different time points; however, the same underlying principle with comparable/the same controls were used to control for possible assay drift.

Cardiometabolic risk was defined as the presence of more than two parameters of MetS, using the international criteria of fasting plasma glucose ≥5.6 mmol/L or oral hypoglycaemic drug treatment, systolic blood pressure (SBP) ≥130 and/or diastolic blood pressure (DBP) ≥85 mm Hg, or antihypertensive drug treatment, serum TG ≥1.7 mmol/L, HDL-cholesterol ≤ 1 mmol/L in men, or ≤1.3 mmol/L in women, or hypolipidemic drug treatment, and waist circumference ≥80 cm in women or ≥94 cm in men [16]. An earlier study in the same participants showed increased cardiometabolic risk among women at a BMI of 26 to 28 kg/m^2^ and at a BMI of 21 to 23 kg/m^2^ for men, while a BMI of 30 kg/m^2^ underestimated cardiometabolic risk [7]. Therefore, we applied a cut-off point of 25 kg/m^2^ instead of 30 kg/m^2^ to define MHO and metabolically unhealthy obesity (MUO) in this study.

Nurses used the First Response HIV card test (PMC Medical, Daman, India) according to the South African protocol after counselling and confirmed HIV positive test results with a Pareeshak card test (BHAT Bio-tech, Bengaluru, India) in 2005 and an Abon test (Biopharm Corporation Limited, Hangzhou, China) in 2015. They referred those with a positive HIV test to local clinics for a confirmation tests and appropriate medical follow-up. Individuals identified with any abnormal values were referred to the nearest clinic or hospital.

### 2.3. Ethical Considerations

The Ethics Committee of the North-West University (numbers: 04M10, NWU-00016-10-A1) approved the study. All procedures complied with the Declaration of Helsinki. The Provincial Department of Health, Local Government Authorities and tribal chiefs in rural communities were notified and approved the study. All participants gave written informed consent for the study procedures. They were transported to and from study sites and received meals after blood sampling.

### 2.4. Statistical Methods

Baseline and follow-up data of the study participants were tested for a normal distribution of data by Q–Q plots and the Kolmogorov–Smirnov test. The mean ± standard deviation (SD) for continuous variables with a normal distribution were presented, together with numbers and percentages for categorical variables. The median and interquartile range (IQR) were presented for non-normally distributed continuous data. We compared baseline data of the participants with complete follow-up for 10 years with those who were lost to follow-up using the Mann–Whitney test for continuous data, because most variables had a non-normal distribution, and chi-square tests for categorical variables. The Wilcoxon sign rank test was used to assess differences between the participants’ baseline and 10-year follow-up data. MHO or MUO categories were defined according to BMI category and presence of MetS at baseline and end. The participants in the MHO category had a BMI ≥ 25 kg/m^2^, but not MetS [4,5,16,24], whereas those in the MUO category had a BMI ≥25 kg/m^2^ and MetS. The following main groups were defined: maintenance of a normal BMI plus the absence of MetS in 2005 and 2015 (1 = metabolically healthy normal weight, MHNW), maintenance of MHO in 2005 and 2015 (2 = MHOMHO), transition from MHO in 2005 to MUO in 2015 (3 = MHOMUO) and MUO in 2005 and 2015 (4 = MUOMUO). Data analysis focussed on these four main groups, but frequencies for metabolically unhealthy normal weight (MUNW) were also calculated, as well as the transition from MHNW to MUNW and MUNW to other categories. Differences among the outcome variables from the four main groups of interest were assessed using ANOVA, with post hoc Bonferroni analysis for variables with a normal distribution and Kruskal–Wallis test for variables with a non-normal distribution. When outcome variables with a non-normal distribution were compared between two groups consecutively, the Mann–Whitney test was used (e.g., MHOMHO, MHOMUO and MUOMUO, respectively, compared with MHNW and MHNW vs. MHO, as well as MHO vs. MUO). All statistical analyses were conducted using SPSS statistics for Windows version 27 (IBM, Armonk, NY, USA).

## 3. Results

### 3.1. Baseline and Follow-Up Characteristics of the Participants

The baseline age, waist circumference, blood pressure and serum lipids of those followed up in 2015 and those not available for follow-up did not differ significantly (data not shown). Among those who were followed up in 2015, the proportion of male dropouts was significantly higher than females (43.7% vs. 29.7%). Loss to follow-up was higher among HIV-positive participants (21.6% vs. 11.6%) due to death within the 10-year period. More participants with a high school education (61% vs. 54.4%) were also lost to follow-up, probably because they moved to better employment opportunities in the cities. At baseline, 10.7% of the participants were newly diagnosed as HIV positive; 16.7% received antihypertensive treatment and only 1% used oral hypoglycaemic drugs. The prevalence of HIV-positive participants increased to 18% after 10 years, 30.8% were treated for hypertension and the proportion treated for type 2 diabetes increased to 4.9%.

The baseline and follow-up anthropometric variables, educational status and smoking habits of the participants are shown in Table 1. There was a significant increase in weight in the female group over time, whereas BMI and waist circumference increased significantly in both men and women. In the total group, the combined overweight/obesity prevalence of both men (19.2% to 23.8%, *p* = 0.02) and women (58% to 64.7%, *p* < 0.001) increased significantly from 2005 to 2015.

### 3.2. Comparison of the Metabolic Profiles of the Different Groups

At baseline, almost half of the overweight and obese participants (45% of the women and 57% of the men) were categorized as MHO. From this group, 43% of women and 47% of men maintained their MHO state throughout the follow-up period, while 46% of the women and 43% of men transitioned to the MUO state. The distribution among the groups is shown in Figure 1. There was a general transition from normal BMI to overweight/obese groups, and from metabolically healthy groups to unhealthy groups. The largest proportion of male participants remained in the MHNW group, while the largest proportion of female participants shifted from MHNW at baseline to MUO at the end of the study. Only 4% (*n* = 37) of the study participants with a BMI within the normal range had MetS and were therefore metabolically unhealthy normal weight (MUNW) at baseline. Only 15 of these 37 adults maintained a normal BMI across the 10 years of follow-up, while the rest moved to the MUO group. Five percent of participants moved from the MHNW to the MUNW category. A small proportion (8%) of participants managed to lose weight, or attain better control over blood pressure, fasting glucose or serum lipids over 10 years and therefore moved from an unhealthier to a healthier category (MUO to MHO or MUNW to MHNW). In contrast, threefold more (24.8%) gained weight and/or their metabolic profiles worsened from MHNW and MHO to the MUNW and MUO categories.

Table 2 shows the baseline and follow-up characteristics according to different MHO and MUO categories. The age of participants who were MUO at baseline and maintained this condition was significantly older than the other three groups. The BMI and waist circumference of the group who maintained MHNW were significantly smaller than the corresponding values for the other groups, both at baseline and after 10 years. The MHOMUO and MUOMUO groups had significantly larger waist circumference than the other groups in 2015. The metabolic profiles of the MHNW, MHO and MUO groups are compared in Table 2; differences are highlighted in Figure 2.

When the two metabolically healthy groups were compared (MHNW and MHOMHO), there were no significant differences between the diastolic blood pressure, HbA1c, fasting glucose, insulin and TG of the two groups, but HOMA-IR, CRP, fibrinogen and PAI-1_act_ were higher and HDL-cholesterol was lower in the MHO group than in the MHNW group. An unexpected result was the higher SBP in the MHNW group compared to the MHO group. Most variables, except HbA1c, fasting glucose, total cholesterol and TG, were not significantly different between the MHOMUO and the MUOMUO groups at follow-up. As expected, the general metabolic profiles (WC, BP, HbA1c, HDL-C, TG, fasting glucose and PAI-1_act_) of the two groups of MUO participants after 10 years (MHOMUO and MUOMUO) were less favourable than those of the groups without MetS. 

## 4. Discussion

The main findings of this study are that about half of the MHO adults maintained their MHO state over 10 years of follow-up, while a similar proportion was no longer metabolically healthy at the end of the study. Although a significantly greater proportion of women than men were overweight/obese at baseline, more men than women could be classified as MHO. These findings have been reported in an earlier study in the same participants [25]. Despite similar metabolic profiles of these MHO adults and the MHNW group in terms of most of the MetS criteria, the HOMA-IR, CRP, fibrinogen and PAI-1_act_ were higher and HDL-cholesterol was lower in the MHOMHO group than the MHNW group, indicating increased cardiometabolic risk in the MHOMHO group [14]. The proportion of men and women with MUO increased considerably over the 10 years, to such an extent that most women were MUO at the end of the study. These findings indicate that MHO is a transient condition and probably age dependent [24,25,26,27], because all three groups that were metabolically healthy at baseline were younger than the MUO group.

Both general obesity and central obesity increased significantly in both men and women. These findings are in line with the global increase in mean BMI among adults since 2000 [1]. Weight gain was associated with the expected unhealthier metabolic profiles associated with obesity [19]. Only a small proportion of participants lost weight or attained improved metabolic profiles over 10 years. The improved metabolic health of these study participants may be due to improved health care with greater access to and availability of treatment for hypertension and type 2 diabetes mellitus [28].

Comparison of metabolically healthy groups (MHNW and MHO) in the present study showed that those who were overweight/obese had significantly lower HDL-cholesterol and higher total cholesterol, HOMA-IR, CRP, fibrinogen and PAI-1_act_ than their counterparts with BMI < 25 kg/m^2^. Obesity, being a chronic low-grade inflammatory condition, is a known driver of a prothrombotic state as a number of proteins involved in haemostasis, including fibrinogen and PAI-1, are acute-phase proteins, which increase during inflammation [29]. Although they did not have the MetS after at least 10 years of being overweight/obese, they were more at risk of developing the MetS and cardiovascular diseases than the leaner group (MHNW). A meta-analysis showed that, compared to the metabolically healthy normal weight groups, participants with MHO were not at an increased risk of all-cause mortality and/or cardiovascular events during a mean follow-up of 11.5 years [27]. Another study showed that healthcare- and loss-of-productivity-related costs were higher among adults with a metabolically unhealthy profile, irrespective of BMI category [24]. However, when studies with follow-up of less than 10 years were excluded from the meta-analysis, the relative risk of cardiovascular event and mortality increased, indicating that MHO may be a transient condition [26]. Although the MHOMHO group in the present study did not develop MetS after at least 10 years of being overweight/obese, they were already at a higher cardiometabolic risk than the MHNW group. The finding that almost half of those who were MHO at baseline transitioned to MUO over the 10 years of follow-up confirms that MHO is a transient condition. The unexpected result of a higher median SBP in the MHNW compared to the MHO group may be due to the fact that smokers were proportionally more represented in the MHNW group than in the MHO group (60.2% vs. 38.4%). The MHNW group also had a higher median daily alcohol intake (2.78 g, IQR 0–22.9 vs. 0g, IQR 0–1) than the MHO group. The positive association between smoking, as well as alcohol intake, with hypertension is well-known in this population [30].

Body fat distribution is regarded a factor that distinguishes MHO from MUO [11]. Both at baseline and after follow-up, there were no differences between the BMI of the groups who maintained MHO over 10 years, and those who maintained MUO across the same period, but the median waist circumference of the MUO group was higher at both intervals. This observation confirms the difference in body composition between the MHO and MUO groups. Furthermore, participants in the group who changed from MHO to MUO over 10 years had a similar median waist circumference at baseline to that of the MHOMHO group, but after follow-up, their median waist circumference increased to be similar to that of the MUOMUO group. Visceral fat is associated with insulin resistance [11,19] and is also a major source of PAI-1. Visceral adipose cells produce pro-inflammatory cytokines, which stimulates PAI-1 secretion (being an acute-phase protein) and contain stromal cells, which is the cellular component of adipose tissue that produces PAI-1 [31,32,33]. In fact, the relationship of PAI-1 with MetS is so strong, it is considered by some to be a true component of the MetS [33]. A recent study confirmed the strong association between obesity, insulin resistance and oxidative stress [34]. Another ten-year observational study in the USA where more than 50% of the participants were Black individuals showed that markers of inflammation were positively associated with incident type 2 diabetes [35].

Although most variables were not significantly different between the MHOMUO and the MUOMUO groups at follow-up, fasting glucose, total cholesterol and TG of the group that maintained MUO over 10 years were significantly higher than those of the group who changed from MHO at baseline to MUO at follow-up. This is an indication that a longer period of living with MUO was associated with a higher fasting glucose and adverse lipid profile compared those who were initially MHO.

A phenotype of normal-weight individuals who are metabolically unhealthy (MUNW) with a higher risk of cardiovascular events or all-cause mortality has also been described and may represent around 20% of the normal weight adult population [36]. Findings from the European Prospective Investigation into Cancer and Nutrition (EPIC) Potsdam study indicate that MUNW participants had higher waist circumference, HbA1c and CRP than their metabolically healthy normal weight counterparts [37]. Two studies provide evidence for the existence of a ‘lipodystrophy-like’ phenotype with a BMI within the normal range [36,37]. The number of normal-weight individuals who were metabolically unhealthy was too small in the present study to compare their metabolic profiles with the other groups.

The limitations of this study include the high loss to follow-up, particularly among men. A large proportion of HIV-positive participants were lost to follow-up because they died early during the 10 years of follow-up. Provincial programmes to treat HIV/AIDS were only initiated in 2005 after initiation of this study and could not prevent early deaths due to HIV/AIDS [38]. More men than women dropped out due to relocation to better job opportunities elsewhere. There is no general agreement about international criteria to define MHO. The definition applied here has been criticized, although many earlier studies used the same definition for MHO, a combination of three or more MetS parameters [16] in combination with a BMI ≥ 25 kg/m^2^ [4,5,24,36]. The strengths of this study include the longitudinal design, which made it possible to track disease progression in adults in an understudied population over a period of ten years. These late adult years are typically the period when cardiovascular risk increases. Several important cardiovascular risk markers were measured in this study.

## 5. Conclusions

An important proportion of the overweight or obese study participants maintained their metabolically healthy state over ten years, but their metabolic profiles were not optimal. The end variables of the MHNW group, in particular HOMA-IR, CRP and clotting factors were more favourable than those of the MHOMHO group. This is an indication that the MHO condition, according to the definition applied here, is transient and not consistent with optimal health. Furthermore, the data highlight the need for the revision and/or standardisation of the definition of MHO. Overweight and obese young adults should therefore not be regarded as healthy in the public health system, but instead should be included in health promotion programmes.

## Figures and Tables

**Figure 1 ijerph-19-05061-f001:**
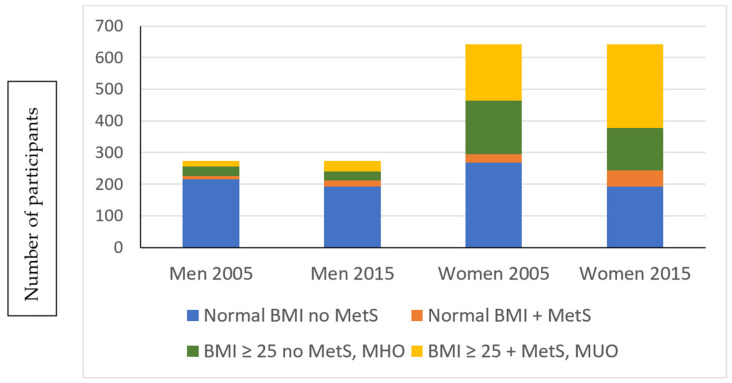
The distribution of participants among the four main BMI and metabolic health (MetS vs. no MetS) groups. BMI = body mass index, MetS = metabolic syndrome, MHO = metabolically healthy overweight/obesity, MUO = metabolically unhealthy overweight/obesity.

**Figure 2 ijerph-19-05061-f002:**
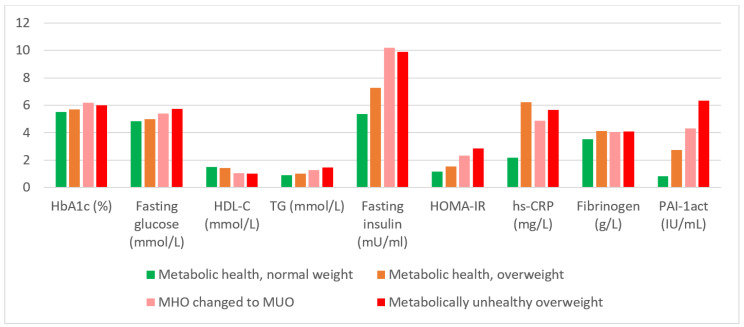
Metabolic profiles of the participants at follow-up according in the four main groups of BMI category and metabolic health.

**Table 1 ijerph-19-05061-t001:** Demographic, anthropometric variables and smoking habits of male and female participants.

Variables	Women (*n* = 642)		Men (*n* = 275)	
	2005	2015	2005	2015
Age (years)	49.3 (43.1–56.4) ^a^	59.2 (53.0–66.6) ^b^	49.8 (42.6–57.7) ^a^	59.7 (52.5–67.9) ^b^
Weight (kg)	63.6 (53.4–78.9) ^a^	66.0 (53.6–81.1) ^b^	57.2 (50.5–66.1) ^a^	57.7 (50.9–67.4) ^a^
Height (m)	1.57 ± 0.06 ^a^	1.56 ± 0.06 ^b^	1.67 ± 0.06 ^a^	1.66 ± 0.07 ^b^
Body mass index (kg/m^2^)	25.9 (21.8–32.0) ^a^	27.0 (22.3–33.4) ^b^	20.1 (18.5–23.4) ^a^	20.6 (18.3–24.3) ^b^
Waist circumference (cm)	80.8 (71.2–91.5) ^a^	91.2 (80.1–102.0) ^b^	75.0 (70.4–82.5) ^a^	78.9 (72.0–91.0) ^b^
Urbanisation:				
Urban area	264 (41.1%)	264 (41.1%)	120 (43.6%)	120 (43.6%)
Rural area	378 (58.9%)	378 (58.9%)	155 (56.4%)	155 (56.4%)
Educational status:				
School education	418 (65.1%)	418 (65.1%)	165 (60%)	165 (60%)
No school education	225 (34.9%)	225 (34.9%)	110 (40%)	110 (40%)
Smoking:				
Smoker	287 (44.7%)	210 (32.7%) ^b^	153 (55.6%) ^a^	136 (49.4%) ^b^
Non-smoking	355 (55.3%)	432 (67.3%)	122 (44.4%)	139 (50.6%)

Non-normal data presented as median (interquartile range). ^a,b^ different superscripts indicate significant differences between variables measured in 2005 and 2015.

**Table 2 ijerph-19-05061-t002:** Characteristics of the study participants at baseline and follow-up according to BMI and metabolic syndrome (MetS) groups (MHNW vs. MHO or MUO).

Baseline Variables	MHNW 2005 and 2015; *n* = 345	MHO 2005 and 2015; *n* = 86	MUO 2005 And 2015; *n* = 145	MHO 2005 to MUO 2015; *n* = 91	*p*
Male/female ratio	182/170 (52%/48%)	14/72 (16%/84%)	13/132 (9%/91%)	13/78 (14%/86%)	
Age (years)	47.5 (42.1, 54.6) ^a^	47.9 (42.4, 54.4) ^a^	52.9 (45.6, 58.1) ^b^	47.4 (40.6, 53.7) ^a^	0.027
Body mass index (kg/m^2^)	19.5 (18.0, 21.7) ^a^	33.0 (26.4, 34.1) ^b^	32.5 (28.6, 37.1) ^b^	30.0 (27.0, 32.6) ^b^	<0.0001
Waist circumference (cm)	70.8 (66.2, 74.9) ^a^	85.4 (78.9, 91.9) ^b^	94.6 (88.6, 101.4) ^c^	89.4 (83.3, 97.8) ^b^	<0.0001
Alcohol intake (g)	2.78 (0, 22.9) ^a^	0 (0, 0.97) ^b^	0 (0, 6.12) ^a^	0 (0, 0) ^b^	<0.0001
HIV status	48 (13.6%)	6 (7.0%)	6 (4.1%)	4 (4.4%)	0.01
Smoking	212 (60.2%)	87 (38.4%)	61 (42.1%)	30 (33.0%)	<0.0001
**Variables after 10 years**					
Variable	MHNW 2005 and 2015; *n* = 345	MHO 2005 and 2015; *n* = 86	MUO 2005 and 2015; *n* = 145	MHO 2005 to MUO 2015; *n* = 91	*p*
Body mass index (kg/m^2^)	19.8 (18.0, 22.1) ^a^	32.7 (28.2, 35.6) ^b^	33.1 (29.1, 37.6) ^b^	32.0 (29.2, 35.8) ^b^	<0.0001
Waist circumference (cm)	75.2 (70.0, 80.0) ^a^	97.7 (91.6, 103.0) ^b^	102.5 (97.3, 110.6) ^c^	101.0 (94.4, 110.0) ^c^	<0.0001
Systolic blood pressure (mm Hg)	127 (112, 147) ^a^	121 (107, 140) ^b^	133 (122, 147) ^c^	133 (122, 150) ^c^	<0.0001
Diastolic blood pressure (mmHg)	82 (74, 94) ^a^	80 (72, 91) ^a^	86 (78, 95) ^b^	91 (83, 96) ^c^	<0.0001
Glycosylated haemoglobin (%)	5.50 (5.2, 5.7) ^a^	5.70 (5.5, 6.0) ^a^	6.0 (5.7, 6.7) ^b^	6.20 (5.7, 6.9) ^c^	<0.0001
Total cholesterol (mmol/L)	4.29 (3.64, 5.04) ^a^	4.68 (4.05, 5.50) ^b^	4.93 (4.22, 5.60) ^c^	4.53 (3.85, 5.48) ^a,b^	<0.0001
HDL cholesterol (mmol/L)	1.52 (1.20, 1.84) ^a^	1.42 (1.11, 1.68) ^b^	1.02 (0.81, 1.18) ^c^	1.07 (0.93, 1.20) ^c^	<0.0001
Serum triglycerides (mmol/L)	0.93 (0.71, 1.17) ^a^	1.04 (0.86, 1.29) ^a^	1.46 (1.1, 2.14) ^b^	1.28 (0.95, 2.04) ^c^	<0.0001
Fasting plasma glucose (mmol/L)	4.85 (4.46, 5.26) ^a^	4.98 (4.70, 5.28) ^a^	5.76 (5.08, 6.89) ^b^	5.39 (4.94, 6.06) ^c^	<0.0001
Fasting plasma insulin (mU/mL)	5.35 (3.10, 10.3) ^a^	7.34 (4.8, 10.5) ^a,b^	9.91 (6.07, 17.2) ^c^	10.2 (6.06, 15.0) ^b,c^	<0.0001
HOMA-Insulin resistance	1.16 (0.66, 2.2) ^a^	1.55 (1.03, 2.19) ^b^	2.85 (1.46, 4.92) ^b,c^	2.34 (1.29, 3.74) ^b^	<0.0001
C-reactive protein (mg/L)	2.2 (0.92, 5.52) ^a^	6.21 (2.35, 9.87) ^b^	5.67 (2.91, 9.74) ^b^	4.89 (2.23, 9.62) ^b^	<0.0001
Total fibrinogen (g/L)	3.52 (3.11, 4.12) ^a^	4.14 (3.65, 4.51) ^b^	4.09 (3.52, 4.60) ^b^	4.06 (3.66, 4.57) ^b^	<0.0001
Plasminogen activator inhibitor-1_act_ (IU/mL)	0.85 (0.00, 4.39) ^a^	2.76 (0.00, 6.19) ^b^	6.34 (1.17, 12.3) ^c^	4.30 (0.74, 12.1) ^c^	<0.0001
HIV status	89 (23.3%)	32 (14.0%)	2 (8.0%)	4 (4.9%)	0.001
Smoking	189 (51.8%)	57 (25.8%)	7 (28.0%)	16 (21.6%)	<0.0001

MHNW = Normal BMI, no MetS; MUNW = Normal BMI with MetS; MHO = metabolically healthy overweight/obesity; MUO = metabolically unhealthy overweight/obesity; ^a,b,c^ different superscripts indicate significant differences between variables of the four groups.

## Data Availability

Data supporting reported results can be obtained on request from the authors.

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
