# Peer review of "The Metabolic Profiles of Metabolically Healthy Obese and Metabolically Unhealthy Obese South African Adults over 10 Years"

_ijerph, 2022, doi:10.3390/ijerph19095061_

Round 1

Reviewer 1 Report

The work written by Kruger et al. concerns a very important issue, because obesity and its complications constitute one of the most important medical problems in the world associated with premature death, worsening of the quality of life as well as burden of healthcare systems. The great value of this study is the comparison of the state of the same population at a 10-year interval. The results presented may contribute to a better understanding of the term “metabolically healthy obesity” elucidating that it is indeed different from heath. The work is generally well prepared and it should be considered for publication. However, I believe that some changes to the text are needed which may add to the value of this manuscript.

The introduction is a bit laconic and should be developed. I believe that it is worth developing the issue of the importance and foundations of the pathogenesis of cardiovascular diseases, which are the main complication of metabolic disorders in the course of obesity. In addition, the issue of oxidative stress, which is of great interest in obesity research, should be mentioned. It is worth mentioning the most important recent achievements in the field of research in the field of these issues. (doi.org/10.1155/2021/9987352; doi.org/10.3390/antiox11010079; doi.org/10.3390/antiox9030236; DOI: 10.3803/EnM.2022.101; doi.org/10.1093/toxsci/kfx233).

It should be “glycated hemoglobin (HbA1c)” to explain the abbreviation HbA1c when used the first time. (line 84)

It should be explained what questions where asked in the mentioned questionnaires, or the whole questionnaires should be available in supplementary materials. If it was described in another publication, it should be mentioned. (lines 85-86)

In the case of numerical values with a unit, there should be a space between the number and the unit. (I found such error in lines 89, 90).

It should be rather “glycated hemoglobin”, not “glycosylated”, because this derivate is the product of nonenzymatic glycation. (lines 108-109)

Should be unified throughout the text, whether there should be spaces or not between the =, > or <, and the numbers.

It is worth noting what new conclusions this study brought in comparison to the aforementioned study on the same population.

The list of references is prepared not in accordance with the rules required in papers published by MDPI. It should be improved according to instructions for Authors.

English seems to be quite good (the text is understandable for me). I’m not English philologist and I feel not fully able to assess language quality.

Author Response

We appreciate the thorough reviews and addressed the comments of the reviewers as follows:

Reviewer 1:

I believe that some changes to the text are needed which may add to the value of this manuscript. The introduction is a bit laconic and should be developed. I believe that it is worth developing the issue of the importance and foundations of the pathogenesis of cardiovascular diseases, which are the main complication of metabolic disorders in the course of obesity. In addition, the issue of oxidative stress, which is of great interest in obesity research, should be mentioned. It is worth mentioning the most important recent achievements in the field of research in the field of these issues. (doi.org/10.1155/2021/9987352; doi.org/10.3390/antiox11010079; doi.org/10.3390/antiox9030236; DOI: 10.3803/EnM.2022.101; doi.org/10.1093/toxsci/kfx233).

Thank you for suggesting references to develop the issue of the foundations of the pathogenesis of cardiovascular diseases, in particular oxidative stress. We considered all references, but we cited only paper doi.org/10.3390/antiox9030236 in a sentence added in the Introduction (lines 90-92) and paper doi.org/10.3390/antiox11010079 in the Discussion (lines 337-340), rather than the Introduction.

We did not use the following based on reasons provided here:

Paper 10.1155/2021/9987352 included only16 MHO (7 women and 9 men) and 61 MUO people and they were all in the 18-36 years age group, whereas all study participants for this manuscript were older than 30 years at baseline (median 49y). Differences between groups were assessed separately for men and women. We believe sample sizes of 7 and 9 in groups were too small to show true differences. Although the total superoxide dismutase (SOD) activity was statistically significantly lower in MUO compared to MHNW in both women and men, no differences between any of the parameters of oxidative stress were found between the MHO and MUO groups.

We believe the importance of the pathogenesis of cardiovascular diseases, which are the main complication of metabolic disorders in the course of obesity should rather be addressed in the Discussion, because the Introduction focuses on the context, what is known and identifying gaps in the current literature, in order to justify the purpose of this study.

Doi.org/10.3390/antiox9030236; We cited this paper in a sentence added in the Introduction: A recent study among older adults revealed that those with the MetS had higher levels of oxidative stress markers, that may contribute to increased inflammation and development of cardiovascular diseases.

DOI: 10.3803/EnM.2022.101 is a review. We prefer to refer mainly to primary studies.

Doi.org/10.1093/toxsci/kfx233) reports on the substance and outcomes of this workshop in 2017 and therefore not a recent primary study.

We mentioned the most important recent achievements in the field of research from these papers:

Paper doi.org/10.3390/antiox11010079:  We included reference to this paper and another in the Discussion, lines 337-340: A recent study confirmed the strong association between obesity, insulin resistance and oxidative stress [Jakubiac] Another ten-year observational study in the USA where more than 50% of the participants were Black individuals showed that markers of inflammation were positively associated with incident type 2 diabetes [Odegaard).

It should be “glycated hemoglobin (HbA1c)” to explain the abbreviation HbA1c when used the first time. (line 84)

Corrected line 110

It should be explained what questions where asked in the mentioned questionnaires, or the whole questionnaires should be available in supplementary materials. If it was described in another publication, it should be mentioned. (lines 85-86)

Added in lines 113-4: Responses included age, educational status, living area, smoking, alcohol consumption, chronic diseases, including HIV and chronic medication used.

In the case of numerical values with a unit, there should be a space between the number and the unit. (I found such error in lines 89, 90).

Corrected throughout

It should be rather “glycated hemoglobin”, not “glycosylated”, because this derivate is the product of nonenzymatic glycation. (lines 108-109)

Corrected line 137

Should be unified throughout the text, whether there should be spaces or not between the =, > or <, and the numbers.

We added spaces throughout consistently.

It is worth noting what new conclusions this study brought in comparison to the aforementioned study on the same population.

The aforementioned study on the same population aimed mainly to identify factors associated with the transition from MHO to MUO and therefore the conclusions are different. The conclusion from both studies was that MHO is a transient stage, but this study added that the metabolic profiles of the MHO were not optimal. The end variables of the MHNW group, in particular HOMA-IR, CRP and clotting factors were more favourable than those of the MHOMHO group.

The list of references is prepared not in accordance with the rules required in papers published by MDPI. It should be improved according to instructions for Authors.

English seems to be quite good (the text is understandable for me). I’m not English philologist and I feel not fully able to assess language quality.

We revised the references in accordance with the rules required in papers published by MDPI.

Our reference style does not allow for CrossRef, but we added the doi number for all papers.

Reviewer 2 Report

I recommend to accept the manuscript after minor revision.

There are only some points to correct:

 - please provide the list of abbreviations

 - please provide the number of ethical approval

- introduction and discussion section need improvement; please provide information on how your results will translate into clinical practice – ex.g. doi: 10.1039/d0fo01878c ; 10.3390/jcm9020469

- in discussion section please provide study strong points  and study limitation section

- please correct typos

All abovementioned issues are crucial for the credibility of the results. The paper can be accepted only after addressing all the issues and another subsequent review.

I recommend to accept the manuscript after minor revision.

Author Response

There are only some points to correct:

 - please provide the list of abbreviations

Added

 - please provide the number of ethical approval

This is provided under 2.3 and in the Institutional Review Board Statement in lines 167-9:

2.3. Ethical considerations

The Ethics Committee of the North-West University (numbers: 04M10, NWU-00016-10-A1) approved the study.

- introduction and discussion section need improvement; please provide information on how your results will translate into clinical practice – ex.g. doi: 10.1039/d0fo01878c; 10.3390/jcm9020469

We added information on how the results will translate into public health practice (lines 376-9), but did not refer to the papers referred to, because these papers addressed different research questions (The effect of Plantago major supplementation on leptin and association of single nucleotide polymorphisms on serum leptin).

- in discussion section please provide study strong points and study limitation section

The limitations were addressed in the last paragraph of the Discussion. We also added strengths there: Strengths of this study include the longitudinal design, which made it possible to track disease progression in adults in an understudied population over a period of ten years. These late adult years are typically the period when cardiovascular risk increases. Several important cardiovascular risk markers were measured in this study (lines 365-9).

- please correct typos

Corrected

Reviewer 3 Report

Global evaluation

This is an interesting paper with a significant public health interest. However, the main aim of this study is not clearly stated. The authors should explain why is it so relevant to observe that MHO is a transient stage to cardiometabolic morbidity. Like this, without implications of this "discovery", this overall observation seems a bit trivial. The paper would be improved with a better contextualization and argumentation in both introduction and discussion on the interest to detect earlier cardiometabolic risks at a transient stage of weight gain. 

Abstract

- The main aim of this study is not clear. Why is it interesting to "determine what proportion of the group maintained MHO over 14-10 years" (lines 14-15)? What is the objective through this statement? 

- If the main aim is to assess the morbid levels along the weight gain spectrum, the authors should better introduce at the beginning of the abstract the context justifying the necessity to address such a goal.

- The conclusion of the abstract appears quite trivial with this formulation (lines 24-25). Why not writing: "Although the metabolic profiles of the MUO group were less favourable than those of their counterparts, MHO is a transient state and associated with increased cardiometabolic risk"?

Introduction

- For the first sentence of this section, I suggest: "...with 39.2% of women and 10% of men being obese".

- Line 37: I suggest to replace "differs across ethnicities and BMI may underestimate" with "differs across ethnicities, so that BMI may underestimate".

- Lines 50-51: There is a problem with this statement:"high blood pressure, fasting plasma glucose, triglycerides and waist circumference and low HDL-cholesterol". There is a mix here between MetS criteria (e.g. high blood pressure, low HDL-cholesterol) and MetS measures (e.g. fasting plasma glucose, waist circumference). You should choose between both.  

- The references 17 and 18 are quite old, around 15 years ago. Can you find more recent papers here?

- As highlighted in the abstract, the main aim of this study is not very clear. Your goal is to address (at least partially) this issue developed in the introduction: "Scientists do not agree on an internationally accepted definition for MHO and the prevalence of MHO varies in different settings"? I think it misses a sentence at the end of the introduction to state clearly what is the goal of this study concerning actual knowledge on MHO.

Materials and Methods

- Why have you used in most cases different methods in 2005 and 2015; e.g. "Fibrinogen was measured using a modified Clauss method (Multifibrin U-test, BCS analyser, Dade Behring, Deerfield, IL, USA) for the 2005 samples and an ACL-200 (Automated Coagulation Laboratory analyser, Instrumentation Laboratories, Milan, Italy) for the 2015 samples"?

- I think there is a problem in this sentence, because you repeat twice the same definition for MHO and MUO: "The participants in the MHO category had a BMI ≥25 kg/m2 and MetS [4,5,16,23], whereas those in the MUO cat egory had a BMI ≥25 kg/m2 and the MetS" (lines 155-157).

Results

- I think that the mention: "in 2005 and 2015." between the Table 1 and its title is useless.

- The Table 1 is difficult to read, especially the last lines. For instance, school education subtitle is not on the same line with its respective values. Same for smoker subtitle. You should make some spaces between variables, particularly between categorical variables (i.e. from urban/rural area to smoker/non smoking).

- What is the unit of the figure 1? %? It is unclear like this without indication.

- In the Table 2, it is not necessary to put: "Variable Baseline variables", you can just put: "Baseline variables".

- The Figure 2 is very interesting. Why is it relevant to compare the metabolic profile of MHO, MUO and other groups? To observe which and how all MetS - and other cardiometabolic - variables are specifically concerned by weight gain spectrum? This to better understand how body fat can affect different metabolic components in order to prevent more efficiently its morbid effects on organism (e.g. adpat BMI standards, develop preventive treatment for cardiometabolic risks associated with the MHO transient stage?, etc.)? And the authors aim to compare these relationships in their population of interest with other studies focusing on the same public health research field? These questions can help you to better state the main aim of this study...

Discussion

- Lines 267-268: "an" instead of "and" here: "greater access to and availability"?

- Lines 300-301: When you wrote: "Furthermore the group who changed from MHO to MUO over 10 years had a similar median waist circumference at baseline to that of the MHO group", the MHO group is the MHOMHO one? Please, detail this.

- Line 319: "had higher a waist circumference" or "higher waist circumference"?

- Lines 320-321: "Both studies provide evidence" or "Two studies provide evidence"? Because you only cited one study until this sentence in this paragraph.

- I think that the authors should more develop the theoretical framework of the discussion, especially by including case studies in other populations focusing on the same research topic. This will help to better compare your findings with those from these works.

Author Response

This is an interesting paper with a significant public health interest. However, the main aim of this study is not clearly stated. The authors should explain why is it so relevant to observe that MHO is a transient stage to cardiometabolic morbidity. Like this, without implications of this "discovery", this overall observation seems a bit trivial. The paper would be improved with a better contextualization and argumentation in both introduction and discussion on the interest to detect earlier cardiometabolic risks at a transient stage of weight gain. 

It is relevant to observe that MHO is a transient and age-dependent stage, because this means that MHO cannot be sustained by around 50% of adults. Such observations can only be made based on data obtained from longitudinal studies.

Furthermore, the results show that despite similar metabolic profiles of these MHOMHO adults and the MHNW group in terms of most MetS criteria, the HOMA-IR, CRP, fibrinogen and PAI-1act were higher and HDL-cholesterol was lower in the MHOMHO than the MHNW group, indicating increased cardiometabolic risk in the MHOMHO group (lines 282-283). This shows that the MHOMHO group is not metabolically healthy (higher HOMA-IR, CRP, fibrinogen and PAI-1act and lower HDL-cholesterol than MHNW group) and therefore should not be regarded as healthy in the public health system.

We revised the Introduction and Discussion to improve contextualization and argumentation regarding the interest to detect earlier cardiometabolic risks at a transient stage of weight gain. We added the following sentence at the end of the Conclusion, lines 376-9:

This data furthermore highlights the need for the revision and/or standardisation of the definition of MHO Overweight and obese young adults should therefore not be regarded as healthy in the public health system but should be included in health promotion programmes.

Abstract

- The main aim of this study is not clear. Why is it interesting to "determine what proportion of the group maintained MHO over 14-10 years" (lines 14-15)? What is the objective through this statement? 

We believe it is important to keep this part of the main aim, therefore, to first determine what proportion of the group maintained MHO over 10 years, before comparing the metabolic profiles of the metabolically healthy and metabolically unhealthy groups after the follow-up period. In our opinion, and taking the controversy around the concept of MHO into account, we believe that it is interesting to determine what proportion of the group maintained MHO over 10 years. Lines 14-15 are left unchanged.

- If the main aim is to assess the morbid levels along the weight gain spectrum, the authors should better introduce at the beginning of the abstract the context justifying the necessity to address such a goal.

The main aim is not to assess the morbid levels along the weight gain spectrum, but to compare the metabolic profiles of the metabolically healthy and metabolically unhealthy groups after the 10-year follow-up period, in order to assess if the metabolically healthy group is really metabolically healthy compared to the metabolically unhealthy group.

- The conclusion of the abstract appears quite trivial with this formulation (lines 24-25). Why not writing: "Although the metabolic profiles of the MUO group were less favourable than those of their counterparts, MHO is a transient state and associated with increased cardiometabolic risk"?

Changed as suggested, lines 23-5.

Introduction

- For the first sentence of this section, I suggest: "...with 39.2% of women and 10% of men being obese".

Changed as suggested line 52.

- Line 37: I suggest to replace "differs across ethnicities and BMI may underestimate" with "differs across ethnicities, so that BMI may underestimate".

Changed as suggested in line 59.

- Lines 50-51: There is a problem with this statement:"high blood pressure, fasting plasma glucose, triglycerides and waist circumference and low HDL-cholesterol". There is a mix here between MetS criteria (e.g. high blood pressure, low HDL-cholesterol) and MetS measures (e.g. fasting plasma glucose, waist circumference). You should choose between both.  

All 5 are MetS criteria according to the harmonised definition (Alberti et al 2009).

- The references 17 and 18 are quite old, around 15 years ago. Can you find more recent papers here?

We updated reference 17, but believe reference 18 is the most appropriate reference for the statement and still relevant.

- As highlighted in the abstract, the main aim of this study is not very clear. Your goal is to address (at least partially) this issue developed in the introduction: "Scientists do not agree on an internationally accepted definition for MHO and the prevalence of MHO varies in different settings"? I think it misses a sentence at the end of the introduction to state clearly what is the goal of this study concerning actual knowledge on MHO.

The goal of this study is not to propose an internationally accepted definition for MHO, but to contribute knowledge that may assist an international working group to establish an internationally accepted definition for MHO. We did not collect an adequate variety of variables in this study to propose an internationally accepted definition for MHO. We added a sentence at the end of the introduction: ‘’These results contribute knowledge that may assist an international working group to establish an internationally accepted definition for MHO, which may need to include more CVD risk markers that MetS alone’’ (lines 90-2).

Materials and Methods

- Why have you used in most cases different methods in 2005 and 2015; e.g. "Fibrinogen was measured using a modified Clauss method (Multifibrin U-test, BCS analyser, Dade Behring, Deerfield, IL, USA) for the 2005 samples and an ACL-200 (Automated Coagulation Laboratory analyser, Instrumentation Laboratories, Milan, Italy) for the 2015 samples"?

Different apparatus was used in most cases in 2005 and 2015, because the laboratories had been upgraded and some of the reagents used in 2005 were no longer available in 2015, although the methods remained similar. We added the following to the Methods section to clarify this:

- I think there is a problem in this sentence, because you repeat twice the same definition for MHO and MUO: "The participants in the MHO category had a BMI ≥25 kg/m2 and MetS [4,5,16,23], whereas those in the MUO category had a BMI ≥25 kg/m2 and the MetS" (lines 155-157).

Corrected: MHO or MUO categories were defined according to BMI category and presence of MetS at baseline and end. The participants in the MHO category had a BMI ≥ 25 kg/m2, but not MetS [4,5,16,23], whereas those in the MUO category had a BMI ≥25 kg/m2 and the MetS (line 186).

Results

- I think that the mention: "in 2005 and 2015." between the Table 1 and its title is useless.

Deleted

- The Table 1 is difficult to read, especially the last lines. For instance, school education subtitle is not on the same line with its respective values. Same for smoker subtitle. You should make some spaces between variables, particularly between categorical variables (i.e. from urban/rural area to smoker/non smoking).

Spaces added

- What is the unit of the figure 1? %? It is unclear like this without indication.

Unit added: number of participants

- In the Table 2, it is not necessary to put: "Variable Baseline variables", you can just put: "Baseline variables".

Deleted ‘variables’

- The Figure 2 is very interesting. Why is it relevant to compare the metabolic profile of MHO, MUO and other groups? To observe which and how all MetS - and other cardiometabolic - variables are specifically concerned by weight gain spectrum? This to better understand how body fat can affect different metabolic components in order to prevent more efficiently its morbid effects on organism (e.g. adpat BMI standards, develop preventive treatment for cardiometabolic risks associated with the MHO transient stage? etc.)? And the authors aim to compare these relationships in their population of interest with other studies focusing on the same public health research field? These questions can help you to better state the main aim of this study...

It is relevant to compare the metabolic profile of MHOMHO, MHOMUO, MUOMUO with the MHNW group, to show that despite similar metabolic profiles of these MHOMHO adults and the MHNW group in terms of most MetS criteria, the HOMA-IR, CRP, fibrinogen and PAI-1act were higher and HDL-cholesterol was lower in the MHOMHO than the MHNW group, indicating increased cardiometabolic risk in the MHOMHO group (lines 277-279). This shows that the MHOMHO group is not metabolically healthy (higher HOMA-IR, CRP, fibrinogen and PAI-1act and lower HDL-cholesterol than MHNW group). The variables are not specifically concerned by weight gain spectrum, because some variables were similar between MHNW and MHOMHO groups.

Discussion

- Lines 267-268: "an" instead of "and" here: "greater access to and availability"?

Reformulated to clarify in line 296: ‘’… may be due to improved health care with greater access to and availability of treatment for hypertension and type 2 diabetes mellitus.’’

- Lines 300-301: When you wrote: "Furthermore the group who changed from MHO to MUO over 10 years had a similar median waist circumference at baseline to that of the MHO group", the MHO group is the MHOMHO one? Please, detail this.

Yes corrected, line 330.

- Line 319: "had higher a waist circumference" or "higher waist circumference"?

Corrected

- Lines 320-321: "Both studies provide evidence" or "Two studies provide evidence"? Because you only cited one study until this sentence in this paragraph.

Corrected in line 353

- I think that the authors should more develop the theoretical framework of the discussion, especially by including case studies in other populations focusing on the same research topic. This will help to better compare your findings with those from these works.

We did not include any case studies, because this is an epidemiological study, but we discussed our results and compared the findings with similar epidemiological studies (refs 24-37)

Round 2

Reviewer 1 Report

The authors have satisfactorily answered the questions asked. The manuscript has been improved and may be considered for publication. I suggest only some minor revisions.

In the list of abbreviations: if MHO means metabolically healthy obese/obesity, MUO should mean metabolically unhealthy overweight/obese/obesity (it must be decided whether MUO and MHO are related to adjective or noun, or both). In my opinion, the list of abbreviations should be at the end of the text, not at the beginning (after the conclusion and before the author’s contribution).

In my opinion, the division into paragraphs should be revised. Paragraphs are too long and contain too much information, which reduces the legibility of the text.

Should be applied consistently whether numbers are written with Arabic numerals or with words. For example, it should be not “five per cent”, but 5% (line 239).

I believe that linguistic correctness should be checked and possibly corrected by a linguist.

Author Response

Dear Reviewer,

Thank you for your suggestions.

The division into paragraphs should be revised. Paragraphs are too

long and contain too much information, which reduces the legibility of

the text.

/I broke some paragraphs up into shorter paragraphs, where it made sense./

Should be applied consistently whether numbers are written with

Arabic numerals or with words. For example, it should be not “five

percent”, but 5%.

/Does the reviewers really mean we should start a sentence with a number

= 5%?/

/This is not accepted language. No change./

Thank you so much for valuable suggestions.

Best wishes

Authors

Reviewer 3 Report

The paper is better in this format. I just have two more comments:

- The authors wrote this (lines 81-85): "However, these variables are important predictors of increased risk of CVD and a more detailed definition of MHO, including markers of insulin resistance and inflammation was recently proposed [14]. A recent study among older adults revealed that those with the MetS had higher levels of oxidative stress markers, that may contribute to increased inflammation and development of CVD [20]." Does it mean that MHO people are more concerned by other markers than Mets as insulin resistance and inflammation? It is not very clear here with this formulation, and we don't know if the reference 20 includes specifically MHO participants, whilst this is the main target of this paper. 

- Lines 148-149: "The same underlying principle with comparable / the same controls were however, used to control for possible assay drift." I suggest instead: "The same underlying principle with comparable / similar controls were however used to control for possible assay drift."

Author Response

Dear reviewer,

Does it mean that MHO people are more concerned by other markers than

Mets as insulin resistance and inflammation? It is not very clear here

with this formulation, and we don't know if the reference 20 includes

specifically MHO participants, whilst this is the main target of this

paper.

/It is not clear what ''MHO people are more concerned by other markers

than Mets as insulin resistance and inflammation" means./

/The main target of this paper is MHO and MUO participants./

/In the Introduction we only state what is already known, i.e. ''Fasting

insulin, markers of low-grade inflammation and haemostatic variables are

associated with abnormal metabolic profiles, but are not included as

MetS parameters, because these variables are not measured in primary

care settings.''/

. We suggest instead: "The same underlying principle with comparable /

similar controls were however used to control for possible assay drift."

/Corrected./

Thank you for your valuable suggestions.

Authors